# Implicit ReasoNet: Modeling Large-Scale Structured Relationships with Shared Memory

**Yelong Shen**\*, **Po-Sen Huang**\*, **Ming-Wei Chang** , **Jianfeng Gao**
Microsoft Research, Redmond, WA, USA
`{yeshen,pshuang,minchang,jfgao}@microsoft.com`

## Abstract

Recent studies on knowledge base completion, the task of recovering missing relationships based on recorded relations, demonstrate the importance of learning embeddings from multi-step relations. However, due to the size of knowledge bases, learning multi-step relations directly on top of observed instances could be costly. In this paper, we propose Implicit ReasoNets (`IRNs`), which is designed to perform large-scale inference *implicitly* through a search controller and shared memory. Unlike previous work, `IRNs` use training data to *learn* to perform multi-step inference through the shared memory, which is also jointly updated during training. While the inference procedure is not operating on top of observed instances for `IRNs`, our proposed model outperforms all previous approaches on the popular FB15k benchmark by more than 5.7%.

## 1 Introduction

Knowledge bases such as WordNet (Fellbaum, 1998), Freebase (Bollacker et al., 2008), or Yago (Suchanek et al., 2007) contain many real-world facts expressed as triples, e.g., (*Bill Gates*, `FounderOf`, *Microsoft*). These knowledge bases are useful for many downstream applications such as question answering (Berant et al., 2013; Yih et al., 2015) and information extraction (Mintz et al., 2009). However, despite the formidable size of knowledge bases, many important facts are still missing. For example, West et al. (2014) showed that 21% of the 100K most frequent PERSON entities have no recorded nationality in a recent version of Freebase. We seek to infer unknown relations based on the observed triples. Thus, the knowledge base completion (KBC) task has emerged an important open research problem (Nickel et al., 2011).

Neural-network based methods have been very popular for solving the KBC task. Following Bordes et al. (2013), one of the most popular approaches for KBC is to learn vector-space representations of entities and relations during training, and then apply linear or bi-linear operations to infer the missing relations at test time. However, several recent papers demonstrate limitations of prior approaches relying upon vector-space models alone. By themselves, there is no straightforward way to capture the structured relationships between multiple triples adequately (Guu et al., 2015; Toutanova et al., 2016; Lin et al., 2015a). For example, assume that we want to fill in the missing relation for the triple (`Obama`, `NATIONALITY`, ?), a multi-step search procedure might be needed to discover the evidence in the observed triples such as (`Obama`, `BornIn`, `Hawaii`) and (`Hawaii`, `PartOf`, `U.S.A`). To address this issue, Guu et al. (2015); Toutanova et al. (2016); Lin et al. (2015a) propose different approaches of injecting structured information by directly operating on the observed triplets. Unfortunately, due to the size of knowledge bases, these newly proposed approaches suffer from some limitations, as most paths are not informative for inferring missing relations, and it is prohibitive to consider all possible paths during the training time with expressive models.

In this paper, we take a different approach from prior work on KBC by addressing the challenges of performing large-scale inference through the design of *search controller* and *shared memory*. Our inference procedure centers around the *search controller*, which only operates on the shared memory instead of directly manipulating the observed triples in knowledge base. `IRNs` use training data to

---

\*Equal contribution.

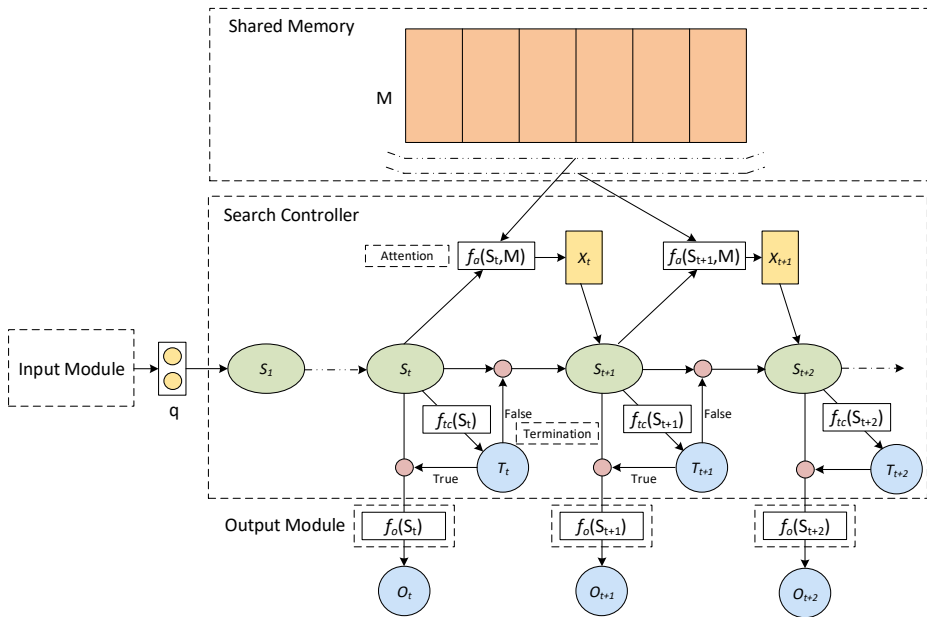

Figure 1: An `IRN` Architecture.

learn to perform multi-step inference through the shared memory. First, *input module* generates a representation of the query. Then, the search controller repeatedly interacts with the *shared memory* and checks the *termination gate*. After each iteration, if the termination condition is met, the model stops the search process and calls the output module to generate a prediction. The shared memory is designed to store key information about the overall structures it learned during training, and hence the search controller only needs to access the shared memory instead of operating on the observed triples.

There are several advantages of using `IRNs`. First, the cost of inference can be controlled because the search controller only needs to access the shared memory. Second, all the modules, including the search controller and memory, are jointly trained, and hence alleviate the needs to inject structured relationships between instances manually. Finally, we can easily extend `IRNs` to other tasks that require modeling structured relationships between instances by switching the input and output modules.

The main contributions of our paper are as follows:

- We propose Implicit ReasoNets (`IRNs`), which use a shared memory guided by a search controller to model large-scale structured relationships implicitly.

- We evaluate `IRNs` and demonstrate that our proposed model achieves the state-of-the-art results on the popular FB15k benchmark, surpassing prior approaches by more than 5.7%.

- We analyze the behavior of `IRNs` for shortest path synthesis. We show that `IRNs` outperform a standard sequence-to-sequence model and execute meaningful multi-step inference.

## 2   REASONET FOR IMPLICIT INFERENCE

In this section, we describe the general architecture of `IRNs` in a way that is agnostic to KBC. `IRNs` are composed of four main components: an input component, an output component, a shared memory, and a search controller, as shown in Figure 1. In this section, we briefly describe each component.

Input/Output Modules: These two modules are task-dependent. The input module takes a query and converts the query into a vector representation $q$. The output module is a function $f_o$, which converts the hidden state received from the search controller ($s$) into an output $O$. We optimize the whole

model using the output prediction $O$ with respect to a ground-truth target using a task-specified loss function.

Shared Memory: The shared memory is denoted as $M$. It consists of a list of memory vectors, $M = \{m_i\}_{i=1...I}$, where $m_i$ is a fixed dimensional vector. The memory vectors are randomly initialized and automatically updated through back-propagation. The shared memory component is shared across all instances.

Search Controller: The search controller is a recurrent neural network and controls the search process by keeping internal state sequences to track the current search process and history. The search controller uses an attention mechanism to fetch information from relevant memory vectors in $M$, and decides if the model should output the prediction or continue to generate the next possible output.

- *Internal State*: The internal state of the search controller is denoted as $S$, which is a vector representation of the search process. The initial state $s_1$ is usually the vector representation of the input vector $q$. The internal state at $t$-th time step is represented by $s_t$. The sequence of internal states is modeled by an RNN: $s_{t+1} = \text{RNN}(s_t, x_t; \theta_s)$.

- *Attention to memory*: The attention vector $x_t$ at $t$-th time step is generated based on the current internal state $s_t$ and the shared memory $M$: $x_t = f_{att}(s_t, M; \theta_x)$. Specifically, the attention score $a_{t,i}$ on a memory vector $m_i$ given a state $s_t$ is computed as $a_{t,i} = \text{softmax}_{i=1,...,|M|} \lambda \cos(W_1 m_i, W_2 s_t)$, where $\lambda$ is set to 10 in our experiments and the weight matrices $W_1$ and $W_2$ are learned during training. The attention vector $x_t$ can be written as $x_t = f_{att}(s_t, M; \theta_x) = \sum_i^{|M|} a_{t,i} m_i$.

- *Termination Control*: The terminate gate produces a stochastic random variable according to the current internal state, $t_t \sim p(\cdot|f_{tc}(s_t; \theta_{tc}))$. $t_t$ is a binary random variable. If $t_t$ is true, the IRN will finish the search process, and the output module will execute at time step $t$; otherwise the IRN will generate the next attention vector $x_{t+1}$ and feed into the state network to update the next internal state $s_{t+1}$. In our experiments, the termination variable is modeled by a logistical regression: $f_{tc}(s_t; \theta_{tc}) = \text{sigmoid}(W_{tc} s_t + b_{tc})$, where the weight matrix $W_{tc}$ and bias vector $b_{tc}$ are learned during training.

Compared IRNs to Memory Networks (MemNN) (Weston et al., 2014; Sukhbaatar et al., 2015; **?**) and Neural Turing Machines (NTM) (Graves et al., 2014; 2016), the biggest difference between our model and the existing frameworks is the search controller and the use of the shared memory. We build upon our previous work (Shen et al., 2016) for using a search controller module to dynamically perform a multi-step inference depending on the complexity of the instance. MemNN and NTM explicitly store inputs (such as graph definition, supporting facts) in the memory. In contrast, in IRNs, we do not explicitly store all the observed inputs in the shared memory. Instead, we directly operate on the shared memory, which modeling the structured relationships implicitly. We randomly initialize the memory and update the memory with respect to task-specific objectives. The idea of exploiting shared memory is proposed by Munkhdalai & Yu (2016) independently. Despite of using the same term, the goal and the operations used by IRNs are different from the one used in Munkhdalai & Yu (2016), as IRNs allow the model to perform multi-step for each instance dynamically.

## 2.1 STOCHASTIC INFERENCE PROCESS

The inference process of an IRN is as follows. First, the model converts a task-dependent input to a vector representation through the input module. Then, the model uses the input representation to initialize the search controller. In every time step, the search controller determines whether the process is finished by sampling from the distribution according to the terminate gate. If the outcome is termination, the output module will generate a task-dependent prediction given the search controller states. If the outcome is continuation, the search controller will move on to the next time step, and create an attention vector based on the current search controller state and the shared memory. Intuitively, we design whole process by mimicking a search procedure that iteratively finds its target through a structure and output its prediction when a satisfying answer is found. The detailed inference process is described in Algorithm 1.

The inference process of an IRN is considered as a Partially Observable Markov Decision Process (POMDP) (Kaelbling et al., 1998) in the reinforcement learning (RL) literature. The IRN produces

---

**Algorithm 1:** Stochastic Inference Process in an `IRN`

---

**Input** : Randomly initialized shared memory $M$; Input vector $q$; Maximum step $T_{\max}$
**Output** : Output vector $o$

1 Define $s_1 = q$; $t = 1$;
2 Sample $t_t$ from the distribution $p(\cdot | f_{tc}(s_t; \theta_{tc}))$;
3 if $t_t$ is false, go to Step 4; otherwise Step 7;
4 Generate an attention vector $x_t = f_{att}(s_t, M; \theta_x)$;
5 Update the internal state $s_{t+1} = \text{RNN}(s_t, x_t; \theta_s)$;
6 Set $t = t + 1$; if $t < T_{\max}$ go to Step 2; otherwise Step 7;
7 Generate output $o_t = f_o(s_t; \theta_o)$;
8 Return $o = o_t$;

---

the output vector $o_T$ at the $T$-th step, which implies termination gate variables $t_{1:T} = (t_1 = 0, t_2 = 0, ..., t_{T-1} = 0, t_T = 1)$, and then takes prediction action $p_T$ according to the probability distribution given $o_T$. Therefore, the `IRN` learns a stochastic policy $\pi((t_{1:T}, p_T) | q; \theta)$ with parameters $\theta$ to get a distribution over termination actions, and over prediction actions. The termination step $T$ varies from instance to instance. The parameters of the `IRN` $\theta$ are given by the parameters of the embedding matrices $W$ for the input/output module, the shared memory $M$, the attention network $\theta_x$, the search controller RNN network $\theta_s$, the output generation network $\theta_o$, and the termination gate network $\theta_{tc}$. The parameters $\theta = \{W, M, \theta_x, \theta_s, \theta_o, \theta_{tc}\}$ are trained to maximize the total expected reward that the `IRN` when interacting with the environment. The expected reward for an instance is defined as:

$$J(\theta) = \mathbb{E}_{\pi(t_{1:T}, p_T; \theta)} \left[ \sum_{t=1}^{T} r_t \right]$$

The reward can only be received at the final termination step when a prediction action $p_T$ is performed. The rewards on intermediate steps are zeros, $\{r_t = 0\}_{t=1...T-1}$.

We employ the approach from our previous work (Shen et al., 2016), REINFORCE (Williams, 1992) based Contrastive Reward method, to maximize the expected reward. The gradient of $J$ can be written as:

$$\nabla_\theta J(\theta) = \sum_{(t_{1:T}, p_T) \in \mathbb{A}^\dagger} \pi(t_{1:T}, p_T; \theta) \left[ \nabla_\theta \log \pi(t_{1:T}, p_T; \theta)(\frac{r_T}{b^i} - 1) \right]$$

where $\mathbb{A}^\dagger$ is all the possible episodes, the baseline $b^i = \sum_{(t_{1:T}, p_T) \in \mathbb{A}^\dagger} \pi(t_{1:T}, p_T; \theta) r_T$ is the expected reward on the $|\mathbb{A}^\dagger|$ episodes for the $i$-th training instance.

## 3 APPLYING IRNS TO KNOWLEDGE BASE COMPLETION

The goal of KBC tasks (Bordes et al., 2013) is to predict a head or a tail entity given the relation type and the other entity, i.e. predicting $h$ given $(?, r, t)$ or predicting $t$ given $(h, r, ?)$, where ? denotes the missing entity. For a KBC task, the input to our model is a subject entity (a head or tail entity) and a relation. The task-dependent input module first extracts the embedding vectors for the entity and relation from an embedding matrix. We then represent the query vector $q$ for an `IRN` as the concatenation of the two vectors. We randomly initialize the shared memory component. At each step, a training triplet is processed through the model by Algorithm 1, where no explicit path information is given. The `IRN` updates the shared memory implicitly with respect to the objective function. For the task dependent output module, we use a nonlinear projection to project the search controller state into an output vector $o$: $f_o(s_t; \theta_o) = \tanh(W_o s_t + b_o)$, where the $W_o$ and $b_o$ are the weight matrix and bias vector, respectively. We define the ground truth target (object) entity embedding as $y$, and use the $L_1$ distance measure between the output $o$ and target entity $y$, namely $d(o, y) = |o - y|_1$. We sample a set of incorrect entity embeddings $N = \{y_i^-\}_{i=1}^{|N|}$ as negative examples. The probability of

selecting a prediction $\hat{y} \in D$ can be approximated as

$$p(\hat{y}|o) = \frac{\exp(-\gamma d(o, \hat{y}))}{\sum_{y_k \in D} \exp(-\gamma d(o, y_k))}$$

where $D = N \cup \{y\}$. We set $|N|$ and $\gamma$ to 20 and 5, respectively, for the experiments on FB15k and WN18 datasets. The `IRN` performs a prediction action $p_T$ on selecting $\hat{y}$ with probability $p(\hat{y}|o)$. We define the reward of the prediction action as one if the ground truth entity is selected, and zero otherwise.

## 4    EXPERIMENTAL RESULTS

In this section, we evaluate the performance of our model on the benchmark FB15k and WN18 datasets for KBC tasks (Bordes et al., 2013). These datasets contain multi-relations between head and tail entities. Given a head entity and a relation, the model produces a ranked list of the entities according to the score of the entity being the tail entity of this triple. To evaluate the ranking, we report **mean rank (MR)**, the mean of rank of the correct entity across the test examples, and **hits@10**, the proportion of correct entities ranked in the top-10 predictions. Lower MR or higher hits@10 indicates a better prediction performance. We follow the evaluation protocol in Bordes et al. (2013) to report filtered results, where negative examples $N$ are removed from the dataset. In this case, we can avoid some negative examples being valid and ranked above the target triplet.

We use the same hyper-parameters of our model for both FB15k and WN18 datasets. Entity embeddings (which are not shared between input and output modules) and relation embedding are both 100-dimensions. We use the input module and output module to encode subject and object entities, respectively. There are 64 memory vectors with 200 dimensions each, initialized by random vectors with unit $L_2$-norm. We use single-layer GRU with 200 cells as the search controller. We set the maximum inference step of the `IRN` to 5. We randomly initialize all model parameters, and use SGD as the training algorithm with mini-batch size of 64. We set the learning rate to a constant number, 0.01. To prevent the model from learning a trivial solution by increasing entity embeddings norms, we follow Bordes et al. (2013) to enforce the $L_2$-norm of the entity embeddings as 1. We use hits@10 as the validation metric for the `IRN`. Following the work (Lin et al., 2015a), we add reverse relations into the training triplet set to increase the training data.

Following Nguyen et al. (2016), we divide the results of previous work into two groups. The first group contains the models that directly optimize a scoring function for the triples in a knowledge base without using extra information. The second group of models make uses of additional information from multi-step relations. For example, RTransE (García-Durán et al., 2015) and PTransE (Lin et al., 2015a) models are extensions of the TransE (Bordes et al., 2013) model by explicitly exploring multi-step relations in the knowledge base to regularize the trained embeddings. The NLFeat model (Toutanova et al., 2015) is a log-linear model that makes use of simple node and link features.

Table 1 presents the experimental results. According to the table, our model significantly outperforms previous baselines, regardless of whether previous approaches use additional information or not. Specifically, on FB15k, the MR of our model surpasses all previous results by 12, and our hit@10 outperforms others by 5.7%. On WN18, the `IRN` obtains the highest hit@10 while maintaining similar MR results compared to previous work.[1]

To better understand the behavior of `IRN`s, we report the results of `IRN`s with different memory sizes and different $T_{max}$ on FB15K in Table 2. We find the performance of `IRN`s increases significantly if the number of inference step increases. Note that an `IRN` with $T_{max} = 1$ is the case that an `IRN` without the shared memory. Interestingly, given $T_{max} = 5$, `IRN`s are not sensitive to memory sizes. In particular, larger memory always improves the MR score, but the best hit@10 is obtained by $|M| = 64$ memory vectors. A possible reason is that the best memory size is determined by the complexity of the tasks.

We analyze hits@10 results on FB15k with respect to the relation categories. Following the evaluation in Bordes et al. (2013), we evaluate the performance in four types of relation: `1-1` if a head entity

---

[1]Nguyen et al. (2016) reported two results on WN18, where the first one is obtained by choosing to optimize hits@10 on the validation set, and second one is obtained by choosing to optimize MR on the validation set. We list both of them in Table 1.

Table 1: The knowledge base completion (link prediction) results on WN18 and FB15k.

| Model | Additional Information | WN18 | | FB15k | |
|---|---|---|---|---|---|
| | | Hits@10 (%) | MR | Hits@10 (%) | MR |
| SE (Bordes et al., 2011) | NO | 80.5 | 985 | 39.8 | 162 |
| Unstructured (Bordes et al., 2014) | NO | 38.2 | 304 | 6.3 | 979 |
| TransE (Bordes et al., 2013) | NO | 89.2 | 251 | 47.1 | 125 |
| TransH (Wang et al., 2014) | NO | 86.7 | 303 | 64.4 | 87 |
| TransR (Lin et al., 2015b) | NO | 92.0 | 225 | 68.7 | 77 |
| CTransR (Lin et al., 2015b) | NO | 92.3 | 218 | 70.2 | 75 |
| KG2E (He et al., 2015) | NO | 93.2 | 348 | 74.0 | 59 |
| TransD (Ji et al., 2015) | NO | 92.2 | 212 | 77.3 | 91 |
| TATEC (García-Durán et al., 2015) | NO | - | - | 76.7 | 58 |
| NTN (Socher et al., 2013) | NO | 66.1 | - | 41.4 | - |
| DISTMULT (Yang et al., 2014) | NO | 94.2 | - | 57.7 | - |
| STransE (Nguyen et al., 2016) | NO | 94.7 (93) | 244 (**206**) | 79.7 | 69 |
| RTransE (García-Durán et al., 2015) | Path | - | - | 76.2 | 50 |
| PTransE (Lin et al., 2015a) | Path | - | - | 84.6 | 58 |
| NLFeat (Toutanova et al., 2015) | Node + Link Features | 94.3 | - | 87.0 | - |
| Random Walk (Wei et al., 2016) | Path | 94.8 | - | 74.7 | - |
| **IRN** | NO | **95.3** | 249 | **92.7** | **38** |

Table 2: The performance of IRNs with different memory sizes and inference steps on FB15K.

| Number of memory vectors | Maximum inference step | FB15k | |
|---|---|---|---|
| | | Hits@10 (%) | MR |
| $|M| = 64$ | $T_{max} = 1$ | 80.7 | 55.7 |
| $|M| = 64$ | $T_{max} = 2$ | 87.4 | 49.2 |
| $|M| = 64$ | $T_{max} = 5$ | **92.7** | 38.0 |
| $|M| = 64$ | $T_{max} = 8$ | 88.8 | **32.9** |
| $|M| = 32$ | $T_{max} = 5$ | 90.1 | 38.7 |
| $|M| = 64$ | $T_{max} = 5$ | **92.7** | 38.0 |
| $|M| = 128$ | $T_{max} = 5$ | 92.2 | 36.1 |
| $|M| = 512$ | $T_{max} = 5$ | 90.0 | 35.3 |
| $|M| = 4096$ | $T_{max} = 5$ | 88.7 | **34.7** |

can appear with at most one tail entity, 1-Many if a head entity can appear with many tail entities, Many-1 if multiple heads can appear with the same tail entity, and Many-Many if multiple head entities can appear with multiple tail entities. The detailed results are shown in Table 3. The IRN significantly improves the hits@10 results in the Many-1 category on predicting the head entity (18.8%), the 1-Many category on predicting the tail entity (16.5%), and the Many-Many category (over 8% in average).

To analyze the behavior of IRNs, we pick some examples for the tail entity prediction in Table 4. Interestingly, we observed that the model can gradually increase the ranking score of the correct tail entity during the inference process.

## 5 ANALYSIS: APPLYING IRNS TO A SHORTEST PATH SYNTHESIS TASK

We construct a synthetic task, shortest path synthesis, to evaluate the inference capability over a shared memory. The motivations of applying our model to this task are as follows. First, we want to evaluate IRNs on another task requiring multi-step inference. Second, we select the *sequence generation* task so that we are able to analyze the inference capability of IRNs in details.

In the shortest path synthesis task, as illustrated in Figure 2, a training instance consists of a start node and an end node (e.g., $215 \rightsquigarrow 493$) of an underlying weighted directed graph that is unknown to models. The output of each instance is the shortest path between the given start and end nodes of the underlying graph (e.g., $215 \rightarrow 101 \rightarrow 493$). Specifically, models can only observe the start-end node

Table 3: Hits@10 (%) in the relation category on FB15k. (M stands for Many)

| Model | Predicting head $h$ | | | | Predicting tail $t$ | | | |
|---|---|---|---|---|---|---|---|---|
| | 1-1 | 1-M | M-1 | M-M | 1-1 | 1-M | M-1 | M-M |
| SE (Bordes et al., 2011) | 35.6 | 62.6 | 17.2 | 37.5 | 34.9 | 14.6 | 68.3 | 41.3 |
| Unstructured (Bordes et al., 2014) | 34.5 | 2.5 | 6.1 | 6.6 | 34.3 | 4.2 | 1.9 | 6.6 |
| TransE (Bordes et al., 2013) | 43.7 | 65.7 | 18.2 | 47.2 | 43.7 | 19.7 | 66.7 | 50.0 |
| TransH (Wang et al., 2014) | 66.8 | 87.6 | 28.7 | 64.5 | 65.5 | 39.8 | 83.3 | 67.2 |
| TransR (Lin et al., 2015b) | 78.8 | 89.2 | 34.1 | 69.2 | 79.2 | 37.4 | 90.4 | 72.1 |
| CTransR (Lin et al., 2015b) | 81.5 | 89.0 | 34.7 | 71.2 | 80.8 | 38.6 | 90.1 | 73.8 |
| KG2E (He et al., 2015) | **92.3** | 94.6 | 66.0 | 69.6 | **92.6** | 67.9 | 94.4 | 73.4 |
| TransD (Ji et al., 2015) | 86.1 | 95.5 | 39.8 | 78.5 | 85.4 | 50.6 | 94.4 | 81.2 |
| TATEC (García-Durán et al., 2015) | 79.3 | 93.2 | 42.3 | 77.2 | 78.5 | 51.5 | 92.7 | 80.7 |
| STransE (Nguyen et al., 2016) | 82.8 | 94.2 | 50.4 | 80.1 | 82.4 | 56.9 | 93.4 | 83.1 |
| PTransE (Lin et al., 2015a) | 91.0 | 92.8 | 60.9 | 83.8 | 91.2 | 74.0 | 88.9 | 86.4 |
| IRN | 87.2 | **96.1** | **84.8** | **92.9** | 86.9 | **90.5** | **95.3** | **94.1** |

Table 4: Test examples in FB15k dataset, given a head entity and a relation, the `IRN` predicts the tail entity with multiple search steps.

| **Input**: (`Dean Koontz`, /PEOPLE/PERSON/PROFESSION) | | | | | |
|---|---|---|---|---|---|
| **Target**: `Film Producer` | | | | | |
| Step | Termination Prob. | Rank | | Predict top-3 entities | |
| 1 | 0.018 | 9 | Author | TV. Director | Songwriter |
| 2 | 0.052 | 7 | Actor | Singer | Songwriter |
| 3 | 0.095 | 4 | Actor | Singer | Songwriter |
| 4 | 0.132 | 4 | Actor | Singer | Songwriter |
| 5 | 0.702 | 3 | Actor | Singer | **Film Producer** |
| **Input**: (`War and Peace`, /FILM/FILM/PRODUCED_BY) | | | | | |
| **Target**: `Carlo Ponti` | | | | | |
| Step | Termination Prob. | Rank | | Predict top-3 entities | |
| 1 | 0.001 | 13 | Scott Rudin | Stephen Woolley | Hal B. Wallis |
| 2 | 5.8E-13 | 7 | Billy Wilder | William Wyler | Elia Kazan |
| 3 | 0.997 | 1 | **Carlo Ponti** | King Vidor | Hal B. Wallis |

pairs as input and their shortest path as output. The whole graph is unknown to the models and the edge weights are not revealed in the training data. At test time, a path sequence is considered correct if it connects the start node and the end node of the underlying graph, and the cost of the predicted path is the same as the optimal path.

Note that the task is very difficult and *cannot* be solved by dynamic programming algorithms since the weights on the edges are not revealed to the algorithms or the models. To recover some of the shortest paths at the test time, the model needs to infer the correct path from the observed instances. For example, assume that we observe two instances in the training data, "$A \leadsto D$: $A \to B \to G \to D$" and "$B \leadsto E$: $B \to C \to E$". In order to answer the shortest path between $A$ and $E$, the model needs to infer that "$A \to B \to C \to E$" is a possible path between $A$ and $E$. If there are multiple possible paths, the model has to decide which one is the shortest one using statistical information.

In the experiments, we construct a graph with 500 nodes and we randomly assign two nodes to form an edge. We split 20,000 instances for training, 10,000 instances for validation, and 10,000 instances for testing. We create the training and testing instances carefully so that the model needs to perform inference to recover the correct path. We describe the details of the graph and data construction parts in the appendix section. A sub-graph of the data is shown in Figure 2.

For the settings of the `IRN`, we switch the output module to a GRU decoder for a sequence generation task. We assign reward $r_T = 1$ if all the prediction symbols are correct and $0$ otherwise. We use a 64-dimensional embedding vector for input symbols, a GRU controller with 128 cells, and a GRU decoder with 128 cells. We set the maximum inference step $T_{\max}$ to 5.

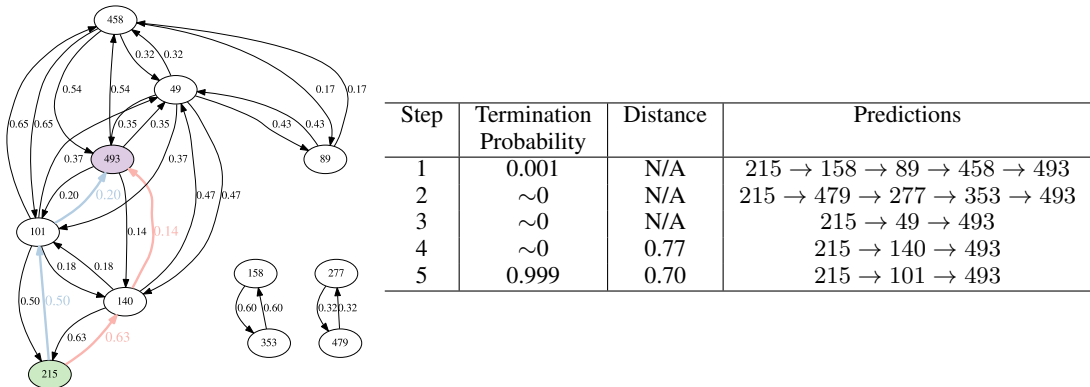

| Step | Termination Probability | Distance | Predictions |
|------|------------------------|----------|-------------|
| 1 | 0.001 | N/A | $215 \rightarrow 158 \rightarrow 89 \rightarrow 458 \rightarrow 493$ |
| 2 | $\sim 0$ | N/A | $215 \rightarrow 479 \rightarrow 277 \rightarrow 353 \rightarrow 493$ |
| 3 | $\sim 0$ | N/A | $215 \rightarrow 49 \rightarrow 493$ |
| 4 | $\sim 0$ | 0.77 | $215 \rightarrow 140 \rightarrow 493$ |
| 5 | 0.999 | 0.70 | $215 \rightarrow 101 \rightarrow 493$ |

Figure 2: An example of the shortest path synthesis dataset, given an input "$215 \rightsquigarrow 493$" (Answer: $215 \rightarrow 101 \rightarrow 493$). Note that we only show the nodes that are related to this example here. The corresponding termination probability and prediction results are shown in the table. The model terminates at step 5.

We compare the `IRN` with two baseline approaches: dynamic programming without edge-weight information and a standard sequence-to-sequence model (Sutskever et al., 2014) using a similar parameter size to our model. Without knowing the edge weights, dynamic programming only recovers 589 correct paths at test time. The sequence-to-sequence model recovers 904 correct paths. The `IRN` outperforms both baselines, recovering 1,319 paths. Furthermore, 76.9% of the predicted paths from `IRN` are *valid* paths, where a path is valid if the path connects the start and end node nodes of the underlying graph. In contrast, only 69.1% of the predicted paths from the sequence-to-sequence model are valid.

To further understand the inference process of the `IRN`, Figure 2 shows the inference process of a test instance. Interestingly, to make the correct prediction on this instance, the model has to perform a fairly complicated inference.[2] We observe that the model cannot find a connected path in the first three steps. Finally, the model finds a valid path at the forth step and predict the correct shortest path sequence at the fifth step.

# 6 RELATED WORK

**Link Prediction and Knowledge Base Completion** Given that $r$ is a relation, $h$ is the head entity, and $t$ is the tail entity, most of the embedding models for link prediction focus on finding the scoring function $f_r(h, t)$ that represents the implausibility of a triple. (Bordes et al., 2011; 2014; 2013; Wang et al., 2014; Ji et al., 2015; Nguyen et al., 2016). In many studies, the scoring function $f_r(h, t)$ is linear or bi-linear. For example, in TransE (Bordes et al., 2013), the function is implemented as $f_r(h, t) = \|\mathbf{h} + \mathbf{r} - \mathbf{t}\|$, where $\mathbf{h}$, $\mathbf{r}$ and $\mathbf{t}$ are the corresponding vector representations.

Recently, different studies (Guu et al., 2015; Lin et al., 2015a; Toutanova et al., 2016) demonstrate the importance for models to also learn from multi-step relations. Learning from multi-step relations injects the structured relationships between triples into the model. However, this also poses a technical challenge of considering exponential numbers of multi-step relationships. Prior approaches address this issue by designing path-mining algorithms (Lin et al., 2015a) or considering all possible paths using a dynamic programming algorithm with the restriction of using linear or bi-linear models only (Toutanova et al., 2016). Toutanova & Chen (2015) shows the effectiveness of using simple node and link features that encode structured information on FB15k and WN18. In our work, the `IRN` outperforms prior results and shows that similar information can be captured by the model without explicitly designing features.

---

[2] In the example, to find the right path, the model needs to search over observed instances "$215 \rightsquigarrow 448$: $215 \rightarrow 101 \rightarrow 448$" and "$76 \rightsquigarrow 493$: $76 \rightarrow 308 \rightarrow 101 \rightarrow 493$", and to figure out the distance of "$140 \rightarrow 493$" is longer than "$101 \rightarrow 493$" (there are four shortest paths between $101 \rightarrow 493$ and three shortest paths between $140 \rightarrow 493$ in the training set).

Studies such as (Riedel et al., 2013) show that incorporating textual information can further improve the knowledge base completion tasks. It would be interesting to incorporate the information outside the knowledge bases in our model in the future.

**Neural Frameworks** Sequence-to-sequence models (Sutskever et al., 2014; Cho et al., 2014) have shown to be successful in many applications such as machine translation and conversation modeling (Sordoni et al., 2015). While sequence-to-sequence models are powerful, recent work has shown that the necessity of incorporating an external memory to perform inference in simple algorithmic tasks (Graves et al., 2014; 2016).

# 7 CONCLUSION

In this paper, we propose Implicit ReasoNets (`IRNs`), which perform inference over a shared memory that models large-scale structured relationships implicitly. The inference process is guided by a search controller to access the memory that is shared across instances. We demonstrate and analyze the multi-step inference capability of `IRNs` in the knowledge base completion tasks and a shortest path synthesis task. Our model, without using any explicit knowledge base information in the inference procedure, outperforms all prior approaches on the popular FB15k benchmark by more than 5.7%.

For future work, we aim to further extend `IRNs` in two ways. First, inspired from Ribeiro et al. (2016), we would like to develop techniques to exploit ways to generate human understandable reasoning interpretation from the shared memory. Second, we plan to apply `IRNs` to infer the relationships in unstructured data such as natural language. For example, given a natural language query such as "are rabbits animals?", the model can infer a natural language answer implicitly in the shared memory without performing inference directly on top of huge amount of observed sentences such as "all mammals are animals" and "rabbits are animals". We believe the ability to perform inference implicitly is crucial for modeling large-scale structured relationships.

ACKNOWLEDGMENTS

We thank Scott Wen-Tau Yih, Kristina Toutanova, Jian Tang and Zachary Lipton for their thoughtful feedback and discussions.

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

## A    DETAILS OF THE GRAPH CONSTRUCTION FOR THE SHORTEST PATH SYNTHESIS TASK

We construct the underlying graph as follows: on a three-dimensional unit-sphere, we randomly generate a set of nodes. For each node, we connect its $K$-nearest neighbors and use the euclidean distance between two nodes to construct a graph. We randomly sample two nodes and compute its shortest path if it is connected between these two nodes. Given the fact that all the sub-paths within a shortest path are shortest paths, we incrementally create the dataset and remove the instances which are a sub-path of previously selected paths or are super-set of previous selected paths. In this case, all the shortest paths can not be answered through directly copying from another instance. In addition, all the weights in the graph are hidden and not shown in the training data, which increases the difficulty of the tasks. We set $k = 50$ as a default value.

