# Peer review of "Implicit ReasoNet: Modeling Large-Scale Structured Relationships with Shared Memory"

_ICLR 2017 — rejected_

[Public Comment · Tsendsuren Munkhdalai · 09 Nov 2016]
**Comment on shared memory**

The idea of shared memory in context of memory augmented neural networks is not novel. Neural Semantic Encoders previously introduced shared and multiple memory accesses [1, 2]. Please discuss the connection between Implicit ReasoNet and Neural Semantic Encoders in your manuscript.

Thanks,


Ref:

1. Munkhdalai, Tsendsuren, and Hong Yu. "Neural Semantic Encoders." arXiv preprint arXiv:1607.04315 (2016).
2. Munkhdalai, Tsendsuren, and Hong Yu. "Reasoning with Memory Augmented Neural Networks for Language Comprehension." arXiv preprint arXiv:1610.06454 (2016).

[Public Comment · (anonymous) · 20 Nov 2016]
**comment on the shared memory**

Can you explain in detail how you construct the shared memory for each training sample

[Official Review · AnonReviewer2 · rating 6 · confidence 4 · 17 Dec 2016]

[Summary]
This paper proposes a new way for knowledge base completion which highlights: 1) adopting an implicit shared memory, which makes no assumption about its structure and is completely learned during training; 2) modeling a multi-step search process that can decide when to terminate.

The experimental results on WN18 and FB15k seem pretty good. The authors also perform an analysis on a shortest path synthetic task, and demonstrate that this model is better than standard seq2seq.

The paper is well-written and it is easy to follow.

[Major comments]
I actually do like the idea and am also impressed that this model can work well.
The main concern is that this paper presents too little analysis about how it works and whether it is sensitive to the hyper-parameters, besides that only reporting a final model on WN18 and FB15k.

One key hyper-parameter I believe is the size of shared memory (using 64 for the experiments). I don’t think that this number should be fixed for all tasks, at least it should depend on the KB scale. Could you verify this in your experiments? Would it be even possible to make a memory structure with dynamic size?

The RL setting (stochastic search process) is also one highlight of the paper, but could you demonstrate that how much it does really help? I think it is necessary to compare to the following: remove the termination gate and fix the number of inference steps and see how well the model does? Also show how the performance varies on # of steps?

I appreciate your attempts on the shortest path synthetic task. However, I think it would be much better if you can demonstrate that under a real KB setting. You can still perform the shortest path analysis, but using KB  (e.g., Freebase) entities and relations.

[Minor comments]
I am afraid that the output gate illustrated in Figure 1 is a bit confusing. There should be only one output, depending on when the search process is terminated.

[Official Review · AnonReviewer3 · rating 6 · confidence 4 · 20 Dec 2016]
**Interesting paper**

In this paper, the authors proposed an implicit ResoNet model for knowledge base completion. The proposed model performs inference implicitly by a search controller and shared memory. The proposed approach demonstrates promising results on FB15k benchmark dataset. 

Pros:

- The proposed approach demonstrates strong performance on FB15k dataset. 

- The idea of using shared memory for knowledge base completion is new and interesting. 

- The proposed approach is general and can be applied in various tasks. 

Cons:

- There is no qualitative analysis on the results, and it is hard to see why the proposed approach works on the knowledge-base completion task. 

- The introduction section can be improved. Specifically, the authors should motivate "shared memory" more in the introduction and how it different from existing methods that using "unshared memory" for knowledge base completion. Similarly, the function of search controller is unclear in the introduction section as it is unclear what does search mean in the content of knowledge base completion.  The concept of shared memory and search controller only make sense to me after reading through section 2.

[Official Review · AnonReviewer1 · rating 6 · confidence 4 · 20 Dec 2016]

This paper proposes a method for link prediction on Knowledge Bases. The method contains 2 main innovations: (1) an iterative inference process that allows the model to refine its predictions and (2) a shared memory component. Thanks to these 2 elements, the model introduced in the paper achieved remarkable results on two benchmarks.


The paper is fairly written. The model is interesting and the experimental results are strikingly good. Still, I only rate for a weak accept for the following reasons.

* The main problem with this paper is that there is little explanation of how and why the two new elements aforementioned are leading to such better results. For instance:
  - What are the performance without the shared memory? And when its size is grown? 
  - How does the performance is impacted when one varies Tmax from 1 to 5 (which the chosen value for the experiments I assume)? This gives an indications of how often the termination gate works.
  - It would also be interesting to give the proportion of examples for which the inference is terminated before hitting Tmax.
  - What is the proportion of examples for which the prediction changed along several inference iterations?

* A value of \lambda set to 10 (Section 2) seems to indicate a low temperature for the softmax. Is the attention finally attending mostly at a single cell? How do the softmax activations change with the type of relationships? the entity type?

* FB15k and WN18 are quite old overused benchmarks now. It would be interesting to test on larger conditions.

[Author Response · yelong shen · 22 Dec 2016]
**Report the performance of IRNs with different memory sizes and inference steps on FB15K**

Thanks all reviewers for your great feedback and comments. We have updated the paper to address the major comments for adding analysis in the KB experiments. We add Table 2 and a corresponding paragraph to analyze the performance with different termination steps and memory sizes. (Note that for T_max =1, it is the case where IRNs do not use the shared memory.)  We found the number of times IRNs access the shared memory is critical for the performance, so IRNs cannot achieve the same level of performance without using shared memory. 
In response to reviewer 2, regarding the shortest path experiments on KB and models with dynamic memory size, we will investigate these directions for future work. 
We will update our paper to improve the introduction section and figure 1, and add more analysis on the termination steps.

[Final Decision · Program Chairs · 06 Feb 2017]
**ICLR committee final decision**

This paper develops a new shared memory based model for doing inference in knowledge bases. The work shows strong empirical results, and potentially could be impactful. However the reviewers felt that the work was not completely convincing without more analysis into the mechanisms of the system itself. 
 
 Pros:
 - Quality: The reviewers like the experimental results of this work, praising them as "strikingly good", but giving the caveat the dataset used is now a bit old for this task. 
 
 Mixed:
 - Clarity: Some reviewers found the work to be fairly well-written although there was mixed opinions about the exposition. Details of the model could be better explained, as could the development of the model
 
 Cons:
 - Quality: The main criticism is not feeling that the methodology is motivated for this task. Multiple reviewers claim there is "little analysis about how it works". Or that was "hard to see" how this would help. All reviewers are in agreement, that the paper should explore more deeply what shared memory is adding to this task, and introduce the approach better in this regard.